# Phylogenetic and Comparative Analysis of *Cryptochironomus*, *Demicryptochironomus* and *Harnischia* Inferred from Mitogenomes (Diptera: Chironomidae)

**DOI:** 10.3390/insects15090642

**Published:** 2024-08-26

**Authors:** Wenbin Liu, Chengyan Wang, Jingyuan Wang, Yaning Tang, Wenxuan Pei, Xinyu Ge, Chuncai Yan

**Affiliations:** Tianjin Key Laboratory of Conservation and Utilization of Animal Diversity, Tianjin Normal University, Tianjin 300387, China; skylwb@tjnu.edu.cn (W.L.);

**Keywords:** Chironomidae, taxonomy, phylogeny, *Harnischia* generic complex

## Abstract

**Simple Summary:**

Chironomidae is ecologically an important family within the order Diptera. With the rapid advances in next-generation sequencing technologies, an increasing amount of mitochondrial genome data for chironomids has been published. The phylogenetic relationships between the three relatively large and morphologically similar genera within the *Harnischia* generic complex, namely *Cryptochironomus*, *Demicryptochironomus*, and *Harnischia*, have yet to be thoroughly investigated. The comprehensive mitotic genomes of 11 species within this generic complex were sequenced. Our findings indicate that the monophyly of the genus *Harnischia* is strongly supported across all topologies, revealing that *Cryptochironomus* is a sister taxon to *Demicryptochironomus*. This represents a new insight into the systematic study of the *Harnischia* generic complex.

**Abstract:**

(1) Background: Mitochondrial genomes have been extensively employed as a crucial marker in numerous dipteran families for understanding phylogenetics and systematics relations, thereby playing a pivotal role in molecular biology studies. The phylogenetic relationship of the *Harnischia* generic complex remains contentious due to the paucity of taxonomic and molecular data. Specifically, the evolutionary relationships among *Cryptochironomus*, *Demicryptochironomus*, and Harnischia are still unclear. (2) Methods: In this study, *Polypedilum* and *Endochironomus* were used as outgroups to analyze phylogenetic relationships among *Cryptochironomus*, *Demicryptochironomus*, and *Harnischia*, mitogenomes of four *Cryptochironomus*, two *Demicryptochironomus*, two *Harnischia,* and two *Cladopelma* were newly sequenced. Subsequently, we conducted a thorough analysis of the nucleotide composition, sequence length, and evolutionary rate. (3) Results: All mitogenomes exhibited structural conservation, with all genes consistently arranged in the identical order as that of the ancestral mitogenome. Nucleotide composition varied significantly among different genes, and the control region displayed the highest A + T content. All protein-coding genes undergo rigorous purification selection, with the ATP8 gene exhibiting the most rapid evolutionary rate among them. Utilizing Bayesian Inference (BI) and Maximum Likelihood (ML) methods across various databases, we reconstructed the phylogenetic relationships among the genera within the *Harnischia* generic complex, drawing insights from an analysis of 14 mitochondrial genomes. (4) Conclusions: Our results showed that the monophyly of the genera *Harnischia* was well supported in all topologies; *Cryptochironomus* is sister to *Demicryptochironomus*.

## 1. Introduction

Recently, the insect mitochondrial genomes have garnered increased attention in research, exhibiting a remarkable degree of conservation in their mitogenome structure [1,2,3,4,5]. This conservation is evident in the similar gene arrangement they maintain, which is comparable to that of their ancestral insect counterparts [6,7]. The insect’s typical mitochondrial genome, known as the mitogenome, is a double-stranded circular molecule that ranges in size from 14 to 20 kilobases (kb) [8]. This genome encodes a comprehensive set of genetic components, including 13 protein-coding genes (PCGs), two ribosomal RNAs (rRNAs), 22 transfer RNAs (tRNAs), and a control region (CR) [1]. Furthermore, due to its small genome size, maternal inheritance, low sequence recombination, and rapid evolutionary rate, the insect mitogenome is deemed a highly effective marker for molecular identification and phylogenetic analysis across various taxonomic levels within Diptera [9,10,11,12,13].

Chironomidae is an ecologically important family within the order Diptera, also known as one of the most abundant and diverse groups in freshwater ecosystems and an important indicator for environmental monitoring [14,15,16,17]. At present, the accurate identification of Chironomidae species and the phylogenetic relationships among some genera are still challenging [18]. With the rapid advances in next-generation sequencing technologies, an increasing amount of mitochondrial genome data for Chironomids has been published [19,20,21,22,23,24]. The structural characteristics of the mitochondrial genome offer valuable insights and evidence for morphological classification, significantly contributing to the study of the phylogeny of Chironomidae [25].

The *Harnischia* generic complex, belonging to the subfamily Chironominae, encompasses 20 genera and over 300 species worldwide [26,27,28]. Initially, the earliest species of the *Harnischia* generic complex were categorized into four distinct species groups under the genus *Tendipes*, *T*. (*C*.) *Paracladopelma* species group, *T*. (*C*.) *Harnischia* species group, *T*. (*C*.) *Cryptochironomus* species group and *T*. (*C*.) *Parachironomus* species group [29]. Subsequently, systematic research on Chironomidae in North America led to the promotion of these four groups to two separate genera [30]. Beck and Beck established the *Harnischia* generic complex in 1969, and soon afterward, numerous scholars conducted systematic studies on this complex or its internal genera, leading to the establishment of several genus-level taxonomic units [31,32,33,34,35,36].

Drawing upon morphological characteristics of female adults, the subfamily Chironominae is classified into three tribes, with robust support for the monophyly of the *Harnischia* generic complex, which forms a sister-group relationship with the *Xenochironomus* [36]. Sasa (1989), focusing on specimens from East Asia, further divided the tribe Chironomini into three generic complexes: *Chironomus* generic complex, *Harnischia* generic complex, and *Polypedilum* generic complex [37]. A comprehensive morphological study of Chironominae’s phylogeny, utilizing 119 morphological traits from 62 genera, reaffirmed the monophyly of the *Harnischia* generic complex [38]. Subsequently, Andersen et al. (2017) encoded 119 morphological traits from 70 genera and constructed a phylogenetic tree for Chironominae, further validating the monophyly of the *Harnischia* generic complex [39,40].

Some species within *Harnischia* generic complex are tolerant to organic pollution, serving as indicator species of eutrophication [41]. A few species are pests to rice crops, damage water supply, or can cause mass outbreaks to emerge that impact human life [42]. Within the *Harnischia* generic complex, although a significant portion of species can be precisely categorized, there remains a level of ambiguity regarding the taxonomic positioning and phylogenetic relationships of certain genera, necessitating further validation and verification [36,43,44]. For the immature stages of these three genera (*Cryptochironomus*, *Demicryptochironomus,* and *Harnischia*), their habitats exhibit considerable similarity, predominantly occupying the bottoms of lakes and large rivers, while a minority can be found inhabiting small streams [45]. In terms of larval structure, the number of antennal flagellomeres, the shape of lateral teeth, and the form of the mentum can readily distinguish these three genera [42]. The pupal characteristics of *Cryptochironomus* and *Harnischia* are relatively similar, and both possess the same pattern of armature on the tergites, whereas *Demicryptochironomus* can be easily distinguished from these two genera by its unique features such as the thoracic horns and the pedes spurii B [46]. The adult insects of these three genera are relatively more difficult to distinguish from each other. From the perspective of the genitalia characteristics of male adults, *Cryptochironomus* and *Harnischia* are more similar, whereas in terms of female characteristics, *Cryptochironomus* and *Demicryptochironomus* are more alike [36,47].

The phylogenetic relationships between the three relatively large and morphologically similar genera within the *Harnischia* generic complex, namely *Cryptochironomus*, *Demicryptochironomus*, and *Harnischia*, have yet to be thoroughly investigated. In this study, we have sequenced, assembled, and annotated the mitogenomes of four *Cryptochironomus* species, two *Demicryptochironomus* species, two *Harnischia* species, and two *Cladopelma* species. Additionally, we incorporated four previously published mitogenomes into our analysis to delve deeper into the characteristics of these mitogenomes. Utilizing Bayesian Inference (BI) and Maximum Likelihood (ML) methods across various databases, we reconstructed the phylogenetic relationships among the genera within the *Harnischia* generic complex, drawing insights from an analysis of 14 mitochondrial genomes. Our findings indicate that the monophyly of the genus *Harnischia* is robustly supported in all topological configurations, further clarifying that *Cryptochironomus* is a sister group to *Demicryptochironomus*.

## 2. Materials and Methods

### 2.1. Taxon Sampling and Sequencing

Our analysis incorporated four *Cryptochironomus* species, two *Demicryptochironomus* species, two *Harnischia* species, and two *Cladopelma* species originating from China (Table 1). Furthermore, for comparative mitogenomic analysis and phylogeny reconstruction, we retrieved the mitogenomes of *Microchironomus tabarui* (Sasa, 1987), *Microchironomus tener* (Kieffer, 1918), *Polypedilum yongsanensis* (Ree and Kim, 1981), *Endochironomus albipennis* Meigen, 1830 in our analysis from GenBank [19,20,21]. Drawing upon extensive previous phylogenetic research on Chironomidae, *Polypedilum yongsanensis* (Ree and Kim, 1981), and *Endochironomus albipennis* (Meigen, 1830) were selected to function as outgroups in our analysis. Prior to DNA extraction and morphological examination, all samples were immersed in a solution of 85% to 95% ethanol at a temperature of −20 °C.

For the extraction of total genomic DNA, the Qiagen DNA Blood and Tissue Kit was utilized. The voucher specimens have been deposited at the College of Life Sciences, Tianjin Normal University, located in Tianjin, China (TJNU), for future reference and analysis. All whole genomes were submitted to Berry Genomics, located in Beijing, China, for sequencing. The Truseq Nano DNA HT Sample Preparation Kit from Illumina (San Diego, CA, USA) was utilized to prepare the sequencing libraries. DNA fragments with an insert size of 350 bp were sequenced using the Illumina Nova 6000 platform (PE150, Illumina), employing a paired-end strategy. Following the trimming of raw reads using Trimmomatic, the resulting clean reads were retained for subsequent downstream analysis [48].

### 2.2. Assembly, Annotation and Composition Analyses

The mitogenome sequences were assembled de novo using NOVOPlasty v3.8.3 (Brussels, Belgium) [49], employing the COI gene as the seed sequence and a range of k-mer sizes from 23 to 39 bp to facilitate the mitogenome assembly process. The annotation of the mitogenome was performed following the methodology described previously by Zheng et al. (2020) [50]. The secondary structure of tRNAs was determined using the MITOS 2 WebServer. The annotation of rRNAs and PCGs was manually carried out in Geneious, utilizing the Clustal Omega algorithm [51]. The analysis of nucleotide composition bias and the nucleotide composition of each gene was conducted using SeqKit v0.16.0, developed in Chongqing, China [52]. The mitogenome map was created utilizing the CGView server, accessible at https://cgview.ca/ (accessed on 25 March 2024). The nucleotide composition, codon usage, and relative synonymous codon usage of the mitogenome were determined using MEGA 11 (Temple University, Philadelphia, PA, USA) [53]. The bias in nucleotide composition was quantified using the AT-skew, calculated as (A − T)/(A + T), and the GC-skew, determined as (G − C)/(G + C). Additionally, the synonymous (Ks) and non-synonymous substitution rates (Ka) of 13 PCGs were computed using DnaSP 6.0 [54].

### 2.3. Phylogenetic Analyses

For phylogenetic analysis, 2 rRNAs and 13 PCGs genes were selectively extracted from 14 mitochondrial genomes. To achieve this, MAFFT (Osaka, Japan) was employed to carry out batch alignment of both nucleotide and protein sequences, utilizing the L-INS-I method to eliminate regions of ambiguous alignment. Trimming was conducted utilizing Trimal v1.4.1 (Barcelona, Spain), followed by the execution of phylogenetic analysis. This analysis was based on five distinct data matrices generated by FASconCAT-G v1.04 (Santa Cruz, CA, USA), specifically: (1) PCG: Including all three codon positions of the 13 protein-coding genes (PCGs); (2) PCG_RNA: Encompassing all three codon positions of the 13 PCGs as well as the two ribosomal RNA (rRNAs); (3) PCG12_RNA: Incorporating the first and second codon positions of the 13 PCGs along with the 2 rRNAs; (4) PCG12: Solely focusing on the first and second codon positions of the 13 PCGs; (5) PCG_AA: Utilizing the amino acid sequences derived from the 13 PCGs. To assess the heterogeneity among the various matrices, AliGROOVE v1.06 (Bonn, Germany) was employed, referencing prior works by Katoh et al. (2013), Capella-Gutiérrez et al. (2009), and Kück et al. (2014) [55,56,57]. Subsequently, the maximum likelihood (ML) and Bayesian inference (BI) trees were constructed using IQ-tree v2.0.7 and Phylobayes-MPI v1.8, respectively.

## 3. Results and Discussion

### 3.1. Mitogenomic Organization

The newly obtained sequences exhibited a length range spanning from 15,662 bp in *Cladopelma edwardsi* to 17,642 bp in *Cryptochironomus rostratus*, with the primary source of this variation being the fluctuating size of the control region (CR; range of sizes from 110 bp in *Cryptochironomus rostratus* to 2450 bp in *Endochironomus albipennis*) (Table 2). All newly assembled mitogenomes encompassed a standardized set of genetic components comprising one control region (CR) and 37 genes, including 13 protein-coding genes (PCGs), 22 transfer RNAs (tRNAs, Appendix A), and 2 ribosomal RNAs (rRNAs; Figure 1). Notably, the lengths of most of these newly assembled mitogenomes were comparable to those of previously published Chironomidae mitogenomes. The sequence characteristics of the represented species are illustrated in Figure 1.

### 3.2. Protein-Coding Genes, Codon Usage, and Evolutionary Rates

Across the various species, there was no significant variation in the size of the tRNA, PCGs, and rRNAs. Specifically, the total length of the 13 PCGs in the acquired mitogenomes ranged narrowly from 11,220 to 11,232 base pairs. When we combined and contrasted our findings with published Chironomidae data, a noteworthy trend emerged: the AT content at the third codon positions of the protein-coding genes (PCGs) was significantly elevated compared to the first and second positions (Figure 2). Strikingly, the majority of the 14 mitogenomes exhibited a negative GC-skew in their PCGs, while each of them displayed a negative AT-skew in the same genes, ranging from −0.200 in *P. yongsanensis* to −0.168 in *D*. *spatulatus*. The AT content, expressed as a percentage, spanned from 73.86 in *D*. *spatulatus* to 77.57 in *D*. *minus*, while the GC content ranged from 22.43 in *D*. *minus* to 26.14 in *D*. *spatulatus* (detailed information is presented in Table 2).

All 13 protein-coding genes (PCGs) in the acquired mitogenomes possessed the standard start codon ATN, which closely aligns with the typical insect mitochondrial start codon [18]. However, variations were observed in other genes. Specifically, the *COI* gene in 10 species utilized TTG as its start codon, while one species employed *ATG*. The *ATP8* gene started with ATT in five species and ATC in five species. The *ND1* gene consistently utilized TTG as its start codon in all species. Additionally, the *ND2*, *ND3*, and *ND6* genes consistently started with ATT, while the *ND5* gene uniquely started with GTG in all species (Figure 3) and provides information on these start codons. Regarding stop codons, 13 PCGs primarily used TAA, with exceptions being *COX2* and *ND3,* which had one TAG.

The Ka/Ks value is a commonly employed metric to quantify the rate of sequence evolution under natural selection [20]. Our findings align closely with those reported in other insect species, revealing that the Ka/Ks ratio for all 13 protein-coding genes (PCGs) was consistently below one, spanning a range from 0.025 (*COX1*) to 0.287 (*ATP8*) (Figure 4). The evolutionary rates of these PCGs can be ranked as follows: *ATP8* > *ND6* > *ND5* > *ND4* > *ND2* > *ND1* > *ND3* > *ND4L* > *CYTB* > *APT6* > *COX3* > *COX2* > *COX1*. Notably, our results suggest that in many cases, genes under purifying selection evolved to eliminate deleterious mutations, operating under varying selection pressures. Specifically, the low ω values observed for *COX2* (0.039) and *COX1* (0.025) suggest a stringent selection environment, while the high ω values for *ATP8* (0.287), *ND6* (0.183), and *ND5* (0.110) indicate a relatively relaxed purifying selection pressure (Figure 4).

The lengths of the 14 mitochondrial tRNAs varied from 1483 to 1510 base pairs (bp), with AT content ranging from 78.81% (in *M*. *tabarui*) to 81.56% (in *E. albipennis*). All tRNAs exhibited a positive AT-skew, spanning values from 0.008 to 0.049. In contrast, the GC content ranged from 18.91% (*D*. *spatulatus*) to 21.19% (*M*. *tabarui*), and the GC-skew varied significantly, from 9.51 (*C*. sp.) to 17.33 (*M*. *tener*).

Regarding the rRNA sequences, their lengths ranged from 2171 bp in *C*. *edwardsi* to 2288 bp in *P. yongsanensis*. The AT content remained consistently high, varying from 83.81% to 84.89%. The GC content, on the other hand, ranged from 15.11% to 16.20%. Notably, the GC-skew of all mitogenomes was significantly positive, ranging from 0.325 to 0.412. While most mitogenomes exhibited a negative AT-skew (−0.028 to −0.001), three species, *H*. *angularis*, *C*. *rostratus*, and *C*. *maculus*, displayed positive values, with AT-skews of 0.005, 0.005, and 0.013, respectively. For a more comprehensive overview of these findings, please refer to Table 2.

### 3.3. Phylogenetic Relationships

The analysis of heterogeneity divergence differences provides a window into the similarities existing in mitochondrial gene sequences across distinct species [58]. Notably, owing to the degeneracy of codons, the dataset AA exhibited the least heterogeneity, whereas the PCG12_rRNA dataset displayed a relatively higher degree of heterogeneity (Figure 5). This observation suggests that the mutation rate of the third codon in protein-coding genes (PCG) surpassed that of the first and second codons. Consequently, the positions of the third codons were deemed unsuitable for reconstructing the phylogenetic relationship among the three genera. Furthermore, it is evident that the heterogeneity observed in the outgroup species of *Polypedilum* and *Endochironomus* is notably lower compared to that found in the ingroup.

Both BI and ML analyses of these five datasets concurred in revealing a consistent topological pattern among the phylogenetic trees despite variations in branch lengths and statistical support (Appendix A). Notably, the monophyly of the genus *Harnischia* was robustly supported in all topologies. Furthermore, our data revealed that within this group, *Cryptochironomus* emerged as a sister taxon to *Demicryptochironomus*, providing valuable insights into their evolutionary history (Figure 6).

There is relatively limited and contradictory research on the systematic studies of these three genera [58]. Previously, a TNT (Tree analysis using New Technology) generated tree based on 74 female characteristics supported *Cryptochironomus* and *Demicryptochironomus* as sister groups, with the rebuilt *Harnischia* sister to (*Cryptochironomus* + *Demicryptochironomus*) [58]. Subsequently, a TNT tree was constructed based on the comprehensive morphological characteristics, encompassed features from the larval, pupal, and adult stages, of 119 species belonging to 70 genera within the Chironominae subfamily, revealing that *Harnischia* and *Cryptochironomus* form sister groups, while *Demicryptochironomus* occupies a basal position within the *Harnischia* generic complex [41]. In the phylogenetic analysis of Chironomidae, using molecular data from fragments of *18SrRNA*, *28SrRNA*, *CAD1*, *CAD4,* and *mtCOI* analyzed by mixed-model Bayesian and maximum likelihood inference methods, *Harnischia* and (*Cryptochironomus* + (*Cryptotendipes* + *Parachironomus*)) were found to be sister groups [40].

However, due to the lack of molecular information for *Harnischia* and the inclusion of only one species per genus, the evolutionary relationships among these three genera based on molecular sequences remain unknown at present, not resolving the traditional morphological analysis as whether *Cryptochironomus* and *Harnischia* are sister groups, or *Harnischia* and *Demicryptochironomus* are sister groups. According to traditional morphological analysis, *Cryptochironomus* and *Harnischia* are sister groups, or *Harnischia* and *Demicryptochironomus* are sister groups. Our findings reveal that *Cryptochironomus* is a sister taxon to *Demicryptochironomus* and indicate the monophyly of the genus *Harnischia*. This study holds significant value as it offers the inaugural molecular data, specifically the mitogenome, for a species belonging to the *Harnischia* generic complex. To further refine and accurately assess the phylogenetic relationships within the *Harnischia* generic complex, mitogenomes from additional species spanning a broader range of genera are imperative.

The nucleotide composition of the newly reported mitogenomes exhibited similarity across the samples (Table 2), reflecting the characteristic AT-biased composition that is typical of Chironomidae and other insect lineages. The AT content of the mitochondrial genomes varied significantly, ranging from 76.31% in *M. tabarui* to 80.34% in *C. rostratus* (Figure 2; Table 2). Notably, the CR exhibited the highest AT content, spanning from 93.25% in *D. spatulatus* to 98.18% in *E. albipennis*. In contrast, the AT content in tRNAs and PCGs was relatively lower than that in rRNAs (Table 2). All newly assembled mitogenomes exhibited a negative GC-skew, indicating a bias toward cytosine, while most displayed a positive AT-skew, reflecting an abundance of adenine and thymine. The GC-skew ranged from −0.240 in *P. yongsanensis* to −0.169 in *D. minus*, while the AT-skew spanned from 0.013 in *D. minus* to 0.050 in *P. yongsanensis*. The GC content itself varied from 19.66% in *C. rostratus* to 23.69% in *M. tabarui*, providing further insights into the nucleotide composition of these mitogenomes (Table 2).

## 4. Conclusions

For the first time, we present a new insight into the genomics of *Harnischia* complex genomics by obtaining the comprehensive mitotic genomes of eleven species spanning four genera within the *Harnischia* generic complex, including four species from the genus *Cryptochironomus*, two from *Demicryptochironomus*, two from *Harnischia*, and two from *Cladopelma*. Additionally, these genomes amalgamated with the previously published mitogenomes of two *Microchironomus* species, enabling us to conduct natural phylogenetic analyses. All newly sequenced mitogenomes displayed strikingly similar structural traits and nucleotide compositions, closely aligning with previously published Chironomidae data.

Our study first offers a mitochondrial genomic perspective on the evolutionary history of the *Harnischia* generic complex. Our findings indicate that the monophyly of the genus *Harnischia* is strongly supported across all topologies, revealing that *Cryptochironomus* is a sister taxon to *Demicryptochironomus*, which constitutes a novel insight into the systematics of the *Harnischia* complex.

## Figures and Tables

**Figure 1 insects-15-00642-f001:**
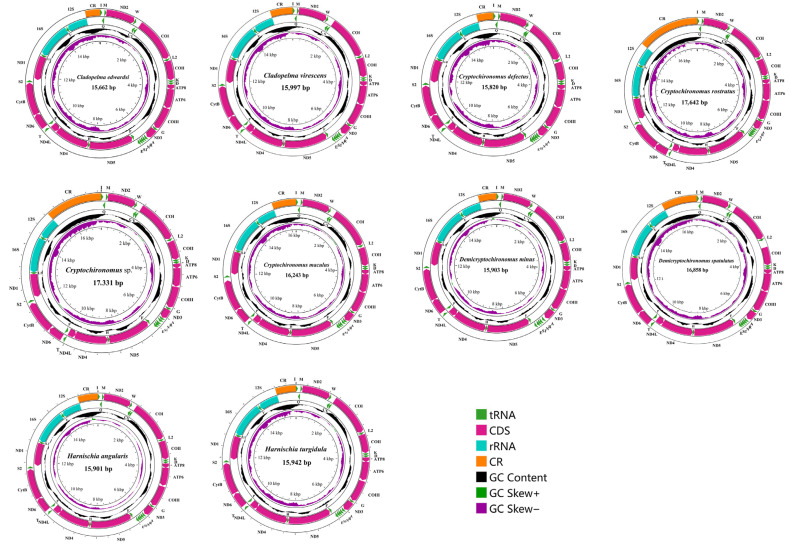
The mitogenome map depicted the distinctive mitochondrial genome attributes of various representative species spanning four genera within the *Harnischia* generic complex. The arrow served as a guide, pointing to the orientation of gene transcription. We adhered to standardized abbreviations to denote Protein-Coding Genes (PCGs) and ribosomal RNAs (rRNAs), while single-letter abbreviations were chosen for transfer RNAs (tRNAs). The second circle highlighted the GC content of the entire mitogenome, whereas the third circle revealed the GC-skew. The innermost circle encapsulated the length of the entire mitogenome, providing a comprehensive visualization of its characteristics.

**Figure 2 insects-15-00642-f002:**
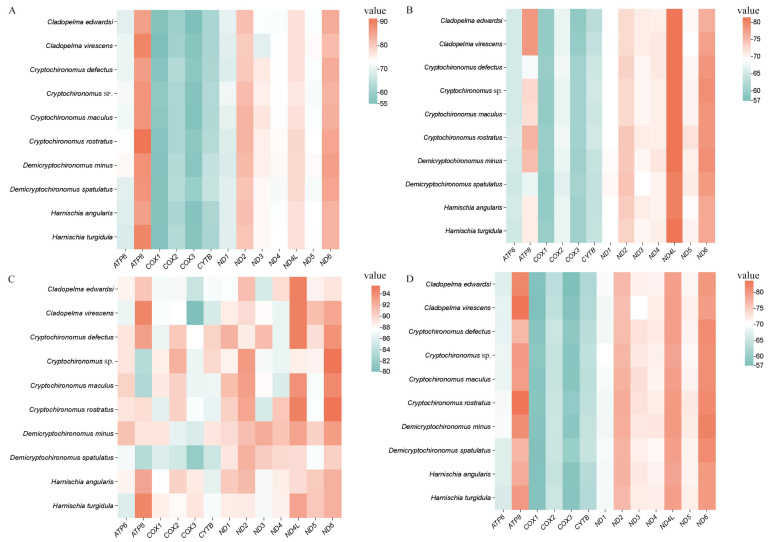
Difference in AT content of protein-coding genes of *Harnischia* generic complex mitogenomes. (**A**) First-codon positions; (**B**) second-codon positions; (**C**) third-codon positions; (**D**) first/second-codon positions.

**Figure 3 insects-15-00642-f003:**
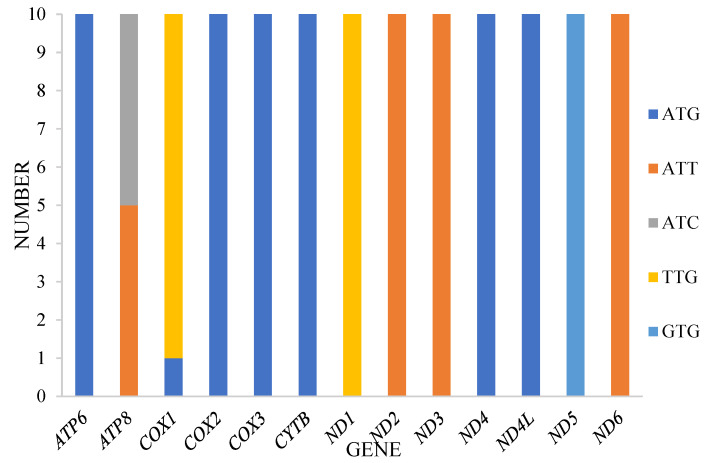
Start codons of protein-coding genes among *Harnischia* generic complex mitogenomes.

**Figure 4 insects-15-00642-f004:**
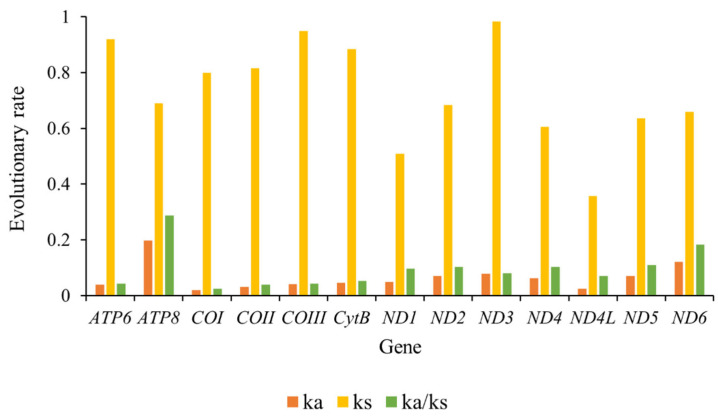
Evolution rate of 13 PCGs of the *Harnischia* generic complex mitogenomes. Ka refers to non-synonymous nucleotide substitutions, Ks refers to synonymous nucleotide substitutions, Ka/Ks refers to the selection pressure of each PCG. The abscissa represented 13 PCGs, and the ordinate represented Ka/Ks values.

**Figure 5 insects-15-00642-f005:**
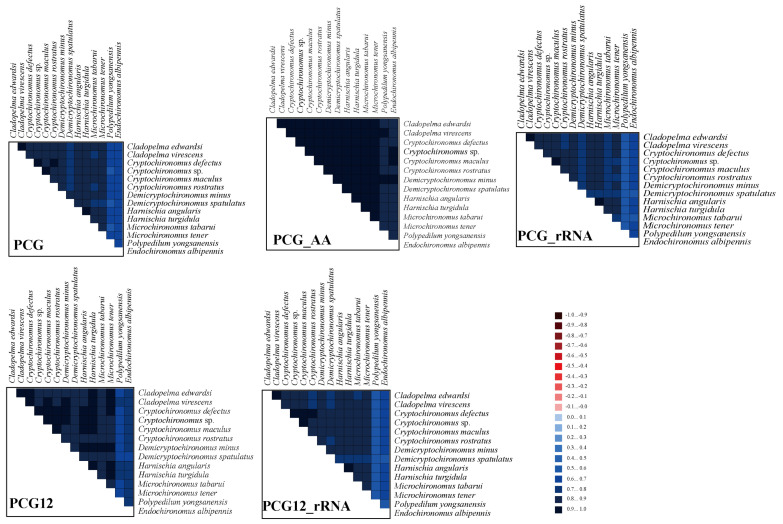
The assessment of the heterogeneity among the mitogenomes of 13 species belonging to the *Harnischia* generic complex focused on their Protein-Coding Genes (PCGs), amino acid sequences, and ribosomal RNAs (rRNAs). The degree of sequence similarity was visually represented through colored blocks, utilizing AliGROOVE scores that span a spectrum from −1 (indicating substantial heterogeneity between datasets, represented by red) to +1 (signaling minimal heterogeneity between datasets, depicted in blue). The lighter hue of each dataset’s-colored block corresponds to a higher degree of heterogeneity, while the darker tone signifies reduced heterogeneity.

**Figure 6 insects-15-00642-f006:**
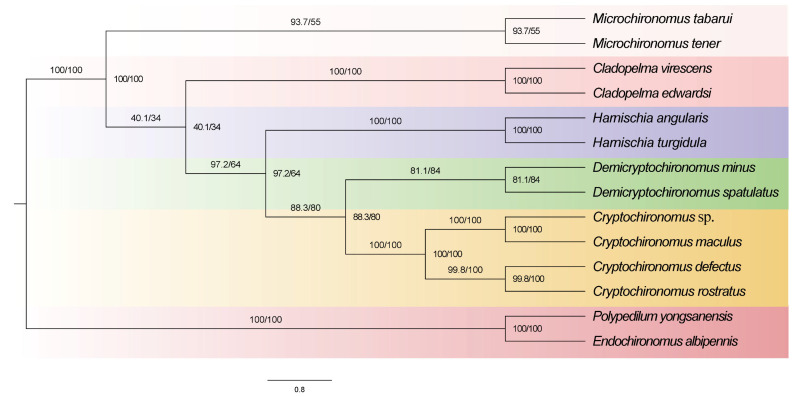
Phylogenetic tree of *Harnischia* generic complex, ML tree based on analysis PCG12_rRNA in partition.

**Table 1 insects-15-00642-t001:** Collection information of newly sequenced species in this study.

Species	Sample ID	Location	Longitude and Latitude	Data	Collector
*Cryptochironomus defectus*	KTM004	Tianmu Mountain, Zhejiang, China	30°31′23″ N; 119°44′59″ E	29.VII.2011	Shu-Li Li
*Cryptochironomus* sp.	KKH002	Kaihua, Zhejiang, China	29°25′00″ N; 118°09′25″ E	8.VII.2017	Wei Cao
*Cryptochironomus maculus*	WGL002	Zunyi, Guizhou, China	28°94′48″ N; 107°88′80″ E	29.VII.2021	Wei Cao
*Cryptochironomus rostratus*	2BW005	Bawangling National Forest Park, Hainan, China	19°08′86″ N; 109°12′92″ E	3.IV.2008	Qian Wang
*Demicryptochironomus minus*	WTS110	Tongren, Guizhou, China	28°31′73″ N; 109°19′95″ E	29.VII.2021	ChunCai Yan
*Demicryptochironomus spatulatus*	KJL001	Kowloon National Wetland Park, Zhejiang, China	28°39′48″ N; 118°84′58″ E	5.VIII.2011	Xiao-Long Lin
*Harnischia angularis*	DTH95	Yueyang, Hunan, China	28°99′21″ N; 112°89′07″ E	30.XII.2021	Fei-Xiang He
*Harnischia turgidula*	KTD484	Taizhou, Zhejiang, China	28°48′67″ N; 121°89′96″ E	20.V.2011	Xing Qi
*Cladopelma edwardsi*	KHZ175	Hangzhou, Zhejiang, China	30°25′51″ N; 120°14′15″ E	23.IV.2011	Xiao-Long Lin
*Cladopelma virescens*	XTC002	Tengchong, Yunnan, China	25°00′67″N; 98°45′82″E	23.IV.2018	Wei Cao

**Table 2 insects-15-00642-t002:** Nucleotide composition of 14 mitogenomes.

Species	Whole Genome					PCG					tRNA				
	Length (bp)	AT%	AT-Skew	GC%	GC-Skew	Length (bp)	AT%	AT-Skew	GC%	GC-Skew	Length (bp)	AT%	AT-Skew	GC%	GC-Skew
*Cladopelma edwardsi*	15,662	77.55	0.036	22.45	−0.211	11,220	76.36	−0.186	23.64	−0.049	1493	79.10	0.016	20.90	14.10
*Cladopelma virescens*	15,997	77.97	0.036	22.03	−0.219	11,220	76.50	−0.184	23.50	−0.070	1483	79.16	0.014	20.84	13.92
*Cryptochironomus defectus*	15,820	78.35	0.022	21.65	−0.185	11,229	77.15	−0.187	22.85	−0.024	1495	80.27	0.023	19.73	15.25
*Cryptochironomus* sp.	17,331	79.85	0.025	20.1	−0.199	11,223	76.90	−0.175	23.10	−0.037	1491	79.54	0.037	20.46	9.51
*Cryptochironomus maculus*	16,243	78.77	0.035	21.23	−0.192	11,220	76.92	−0.173	23.08	−0.050	1494	80.19	0.030	19.81	11.49
*Cryptochironomus rostratus*	17,642	80.34	0.032	19.66	−0.202	11,229	77.54	−0.174	22.46	−0.058	1492	80.63	0.013	19.37	14.88
*Demicryptochironomus minus*	15,903	79.05	0.013	20.95	−0.169	11,223	77.57	−0.177	22.43	−0.037	1498	80.77	0.008	19.23	15.28
*Demicryptochironomus spatulatus*	16,858	78.85	0.016	21.14	−0.235	11,220	76.16	−0.168	23.84	−0.100	1491	81.09	0.049	18.91	12.77
*Harnischia angularis*	15,901	78.42	0.021	21.57	−0.182	11,220	76.66	−0.183	23.33	−0.041	1494	80.12	0.014	19.88	14.48
*Harnischia turgidula*	15,942	78.34	0.028	21.66	−0.195	11,220	76.73	−0.183	23.27	−0.053	1493	80.04	0.011	19.96	13.42
*Microchironomus tabarui*	15,667	76.31	0.042	23.69	−0.203	11,220	74.72	−0.182	25.27	−0.067	1496	78.81	0.030	21.19	11.67
*Microchironomus tener*	15,791	77.73	0.027	22.28	−0.206	11,222	75.64	−0.182	24.36	−0.073	1494	79.92	0.020	20.08	17.33
*Endochironomus albipennis*	15,916	79.80	0.028	20.20	−0.198	11,229	76.93	−0.170	23.07	−0.230	1497	81.56	0.030	18.44	14.49
*Polypedilum yongsanensis*	16,226	77.01	0.050	22.99	−0.240	11,232	73.86	−0.200	26.14	−0.035	1510	80.00	0.030	20.00	14.57
**Species**	**rRNA**					**CR**					**GenBank Accession**			**Reference**	
	**Length (bp)**	**AT%**	**AT-Skew**	**GC%**	**GC-Skew**	**Length (bp)**	**AT%**	**AT-Skew**	**GC%**	**GC-Skew**					
*Cladopelma edwardsi*	2171	83.99	−0.028	16.02	0.380	521	94.24	−0.027	5.76	−0.33	PQ014460			This Study	
*Cladopelma virescens*	2171	83.81	−0.027	16.20	0.393	799	94.75	−0.044	5.13	−0.22	PQ014464			This Study	
*Cryptochironomus defectus*	2200	84.21	−0.003	15.79	0.386	612	95.59	−0.046	4.41	−0.48	PQ014461			This Study	
*Cryptochironomus* sp.	2224	84.74	−0.001	15.27	0.369	2031	94.98	−0.039	4.73	−0.50	PQ014463			This Study	
*Cryptochironomus maculus*	2224	84.43	0.013	15.57	0.386	893	94.85	0.020	5.16	−0.43	PQ014454			This Study	
*Cryptochironomus rostratus*	2208	84.68	0.005	15.34	0.369	2450	94.86	0.034	5.10	−0.20	PQ014455			This Study	
*Demicryptochironomus minus*	2210	84.89	−0.009	15.11	0.325	656	96.34	−0.089	3.66	−0.08	PQ014456			This Study	
*Demicryptochironomus spatulatus*	2252	84.09	−0.009	15.92	0.412	1334	93.25	−0.122	6.60	−0.36	PQ014457			This Study	
*Harnischia angularis*	2208	84.57	0.005	15.44	0.362	741	96.09	−0.048	3.91	−0.17	PQ014458			This Study	
*Harnischia turgidula*	2213	84.21	−0.006	15.79	0.368	739	94.99	−0.046	5.00	−0.24	PQ014459			This Study	
*Microchironomus tabarui*	2182	84.51	−0.020	15.50	0.377	548	94.71	−0.044	5.29	−0.38	MZ261913			[21]	
*Microchironomus tener*	2193	84.25	−0.002	15.76	0.375	674	95.11	−0.020	4.90	−0.45	ON975027			[20]	
*Endochironomus albipennis*	2252	86.01	−0.01	14.00	0.340	110	98.18	−0.100	1.82	1.00	OP950227			[19]	
*Polypedilum yongsanensis*	2288	84.88	−0.02	15.12	0.353	312	95.51	−0.100	4.49	−0.43	OP950222			[19]	

## Data Availability

The following information was supplied regarding the availability of DNA sequences: The new mitogenomes are deposited in GenBank of NCBI and the accession numbers are in Table 2.

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
