# Peer review of "Phylogenetic and Comparative Analysis of *Cryptochironomus*, *Demicryptochironomus* and *Harnischia* Inferred from Mitogenomes (Diptera: Chironomidae)"

_insects, 2024, doi:10.3390/insects15090642_

Round 1

Reviewer 1 Report (New Reviewer)

Comments and Suggestions for Authors

Report on MS 3135236 by Wenbin Liu et al. submitted to MDPI Insects

This manuscript provides new data on the phylogenetic relationships of chironomids within the Harnischiagenus complex based on mitochondrial genome data. By using a set of 14 mitogenomes - 10 newly provided in this study and four from previous studies -, results support the monophyly of the Harnischia group and revealed the existence of sister clades within the group. Both, ML and BI methods for reconstructing the phylogeny showed consistent topologies in the exploration of 15 genes within the mitogenome. All mitogenomes were structurally consistent regarding gene order, but these showed different evolutionary rates. In general, I think the manuscript affords an interesting question as it is to solve the relationship among the genera of the Harnischia complex, the methods implemented seem adequate to me (although I am not an expert in technical aspects of gene annotation) and results help to solve the above-mentioned question.

Nevertheless, I have a few comments that I hope may help in improving the quality of the manuscript. 

First, Since the Harnischia group includes 20 genera, why not to include mitogenomes of each of them. This would definitively help in solving the relationship among the genera of the group. Thus, I ask for a justification of the genera included in the analyses. Why these and not others? Why other genera within the group were discarded?

I am also wondering about the origin of the species and the kind of biological material used. All the species used came from China, but how do results may change if the same species from other locations would be included. This could be a question for further discussion. Also, what specimens were used for DNA extraction? Adults only? Or were larvae and pupae also used?

Other minor comments

l. 53: … an ecologically important family…

l. 89: Which three genera? I guess the authors are referring to HarnischiaCryoptochironomus and Demicryptochironomus. Please clarify here.

l. 104-106: Justify why Cladoplema was included here? I observed Cladoplema was not used as outgroup, so an explanation is needed here.

l. 120: This same question applies to the inclusion of the two Microchironomus species. Do they belong to the Harnischia group too? If so, please state it.

l. 151: Please indicate that Ks and Ka substitution rates were computed for each gene.

l. 222: Figure 2 deserves further explanation. What does each panel show? I guess it is the difference with respect to Microchironomus tabarniM. tenerEndochironomus albipennis and Polypedilum yongsanenis, but no indication of the order is indicated? Please clarify. In addition, a similar scale for the heatmap in each plot would be more helpful.

l. 236 (onwards): I think Ks/Ks is concise enough. There is no need to use w.

l. 291: What is the TNT generated tree?

Author Response

Comments 1: This manuscript provides new data on the phylogenetic relationships of chironomids within the Harnischiagenus complex based on mitochondrial genome data. By using a set of 14 mitogenomes - 10 newly provided in this study and four from previous studies -, results support the monophyly of the Harnischia group and revealed the existence of sister clades within the group. Both, ML and BI methods for reconstructing the phylogeny showed consistent topologies in the exploration of 15 genes within the mitogenome. All mitogenomes were structurally consistent regarding gene order, but these showed different evolutionary rates. In general, I think the manuscript affords an interesting question as it is to solve the relationship among the genera of the Harnischia complex, the methods implemented seem adequate to me (although I am not an expert in technical aspects of gene annotation) and results help to solve the above-mentioned question.

Nevertheless, I have a few comments that I hope may help in improving the quality of the manuscript. 

Response 1: We appreciate very much for your positive and constructive comments and suggestions on our manuscript.

Comments 2: First, Since the Harnischia group includes 20 genera, why not to include mitogenomes of each of them. This would definitively help in solving the relationship among the genera of the group. Thus, I ask for a justification of the genera included in the analyses. Why these and not others? Why other genera within the group were discarded?

Response 2:We agree with this comment. Our article primarily focuses on reporting the mitochondrial genome data of Harnischia cpx and discussing the relationships between three genera that exhibit close morphological similarities and cluster together, rather than exploring the internal relationships within the Harnischia complex.

Comments 3: I am also wondering about the origin of the species and the kind of biological material used. All the species used came from China, but how do results may change if the same species from other locations would be included. This could be a question for further discussion. Also, what specimens were used for DNA extraction? Adults only? Or were larvae and pupae also used?

Response 3:Thanks for your comments. Selecting 2-3 species from each genus can reflect the characteristics of the genus. The materials we selected include larvae and adults.

Comments 4: Other minor comments

  1. 53: … an ecologically important family…

 Response 4:Thanks for your comments. We have made modifications according to your suggestions.

Comments 5: l. 89: Which three genera? I guess the authors are referring to HarnischiaCryoptochironomus and Demicryptochironomus. Please clarify here.

 Response 5: Thanks for your comments. We have made modifications according to your suggestions.

Comments 6: l. 104-106: Justify why Cladoplema was included here? I observed Cladoplema was not used as outgroup, so an explanation is needed here.

Response 6: Adding data from other genera in the Harnischia complex can better understand the relationship between these three genera.

Comments 7: l. 120: This same question applies to the inclusion of the two Microchironomus species. Do they belong to the Harnischia group too? If so, please state it.

 Response 7:Adding data from other genera in the Harnischia  complex can better understand the relationship between these three genera.

Comments 8: l. 151: Please indicate that Ks and Ka substitution rates were computed for each gene.

Response 8:We have added explanations in the MS and marked red.

Comments 9: l. 222: Figure 2 deserves further explanation. What does each panel show? I guess it is the difference with respect to Microchironomus tabarniM. tenerEndochironomus albipennis and Polypedilum yongsanenis, but no indication of the order is indicated? Please clarify. In addition, a similar scale for the heatmap in each plot would be more helpful.

 Response 9:Thanks for your comments. We have revised the image and updated the caption.

Comments 10: l. 236 (onwards): I think Ks/Ks is concise enough. There is no need to use w.

Response 10: Thanks for your comments. We have made modifications according to your suggestions.

Comments 11: l. 291: What is the TNT generated tree?

Response 11:We have added explanations in the MS and marked red.

Reviewer 2 Report (New Reviewer)

Comments and Suggestions for Authors

Dear Authors,

I have reviewed your manuscript with great interest and found it to be well-structured and thoroughly conducted. Below, I provide some suggestions and requests for clarification, mainly concerning the methodological section:

·         Line 37: To enhance the dissemination of your work, I suggest using keywords that differ from those in the title.

·         Lines 119-122: When mentioning the sources from which the mitogenomes were obtained (GenBank [19-21]), it would be helpful to include a brief explanation of why these sources were considered reliable and relevant to your study.

·         Lines 138-139: Why were specific k-mer sizes (23-39 bp) chosen? A brief explanation of these choices could help readers better understand the rationale behind your methodology.

·         Lines 141-143: The section mentions that the annotation of rRNAs and PCGs was performed manually using Geneious, and that Clustal W was employed to verify and refine the boundaries. It might be useful to specify how this manual process was conducted and what criteria were used to determine the final boundaries.

·         I suggest (optional) that you consider including the proposed phylogenetic trees as supplementary material.

·         Please improve the quality of Figure 5, as it is currently unreadable.

Best regards, and good luck with your work.

Comments on the Quality of English Language

Minor editing of English language required.

Author Response

Comments 1: I have reviewed your manuscript with great interest and found it to be well-structured and thoroughly conducted. Below, I provide some suggestions and requests for clarification, mainly concerning the methodological section:

Response 1:We appreciate very much for your positive and constructive comments and suggestions on our manuscript.

Comments 2: Line 37: To enhance the dissemination of your work, I suggest using keywords that differ from those in the title.

Response 2: Thanks for your comments. We have made modifications according to your suggestions.

Comments 3: Lines 119-122: When mentioning the sources from which the mitogenomes were obtained (GenBank [19-21]), it would be helpful to include a brief explanation of why these sources were considered reliable and relevant to your study.

Response 3:We selected two non- Harnischia cpx species with close relationships from the Chironomidae family as outgroups for analysis.

Comments 4: Lines 138-139: Why were specific k-mer sizes (23-39 bp) chosen? A brief explanation of these choices could help readers better understand the rationale behind your methodology.

Response 4:k-mer sizes (23-39 bp) is the default parameter of the software.

Comments 5: Lines 141-143: The section mentions that the annotation of rRNAs and PCGs was performed manually using Geneious, and that Clustal W was employed to verify and refine the boundaries. It might be useful to specify how this manual process was conducted and what criteria were used to determine the final boundaries.

Response 5:Thanks for your comments. We deleted the words that caused ambiguity.

Comments 6: I suggest (optional) that you consider including the proposed phylogenetic trees as supplementary material.

Response 6: Thanks for your comments. Our supporting materials provide phylogenetic trees.

 Comments 7: Please improve the quality of Figure 5, as it is currently unreadable.

Response 7:Thanks for your comments. We have improved the image.

Reviewer 3 Report (New Reviewer)

Comments and Suggestions for Authors

The manuscript presents new mitogenome data for a number of species within the Chironomidae.  The introduction outlines the current understanding of the taxonomy within this group and the need for clarification of the relationships between some taxa. The data presented is largely descriptive but outlines the characteristics of these genomes and the phylogenetic relationships between them. The conclusions are supported by the data which will be useful to researchers interested in this group of insects.

There are a couple of minor issues that need to be addressed:

Line 105 - three Harnischia species were sequenced (should this be two as stated on lines 25 and 118).

Section 3.1 - You outline the size range of the whole genomes and protein coding genes but state that the main difference in size is due to differences in the control region - what is the range of sizes of the CR?

Line 231 -  delete "detailed statistical" (overstating the information in figure 3 a little)

Line 269 - cds12_rrna should be PCG12-RNA to be consistent with how these are described in the methods

Figures 1 and 5 - can the resolution of these figures be improved?

Figure 2 - needs clarification in the figure legend, there are four plots but it unclear what each is showing

Author Response

Comments 1: The manuscript presents new mitogenome data for a number of species within the Chironomidae.  The introduction outlines the current understanding of the taxonomy within this group and the need for clarification of the relationships between some taxa. The data presented is largely descriptive but outlines the characteristics of these genomes and the phylogenetic relationships between them. The conclusions are supported by the data which will be useful to researchers interested in this group of insects.

Response 1:We appreciate very much for your positive and constructive comments and suggestions on our manuscript.

There are a couple of minor issues that need to be addressed:

Comments 2: Line 105 - three Harnischia species were sequenced (should this be two as stated on lines 25 and 118).

Response 2:Thanks for your comments. We have made modifications according to your suggestions.

Comments 3: Section 3.1 - You outline the size range of the whole genomes and protein coding genes but state that the main difference in size is due to differences in the control region - what is the range of sizes of the CR?

Response 3: Thanks for your comments. We have made modifications according to your suggestions.

Comments 4: Line 231 - delete "detailed statistical" (overstating the information in figure 3 a little)

 Response 4:Thanks for your comments. We have made modifications according to your suggestions.

Comments 5: Line 269 - cds12_rrna should be PCG12-RNA to be consistent with how these are described in the methods

Response 5:Thanks for your comments. We have made modifications according to your suggestions.

Comments 6: Figures 1 and 5 - can the resolution of these figures be improved?

Response 6: Thanks for your comments. We have improved the image.

Comments 7: Figure 2 - needs clarification in the figure legend, there are four plots but it unclear what each is showing

 Response 7:Thanks for your comments. We have revised the image and updated the caption.

This manuscript is a resubmission of an earlier submission. The following is a list of the peer review reports and author responses from that submission.

Round 1

Reviewer 1 Report

Comments and Suggestions for Authors

The authors diligently have worked on writing this article, which has yielded results that are not easily obtained, and I commend them. The use of mitogenome for constructing the phylogenetic of Chironomidae species is a new approach, and I am eager to see this study in publication as the information it generates is essential for future studies. However, I would like to raise a point of consideration that may question their entire outcome. The results obtained in this study could potentially be influenced by the outgroup choice, which is from the same generic complex. While I understand that the number of species with full mitogenome sequence is limited, there are still numerous other genera (see Zhang et al. 2023, for instance) that could have been chosen as an outgroup. The authors do not convincingly present an argument for their choice of outgroup. The choice of the outgroup is based on a more distantly related group of organisms, usually in other subfamilies, tribes, or species groups. Please either provide a convincing and comprehensive reason for your choice of the outgroup or redo the phylogenetic analysis with a different outgroup(s). Have the authors done other analyses with different outgroups? If so, please provide those. If choosing other outgroups (e.g., Polypedilum yongsanensis or Endochironomus albipennis or both) over the present outgroups do not change the relationship, then your outgroup choice is still valid.

Comments on the Quality of English Language

In terms of writing, there seems to be a level of haste in getting to the phylogenetic part of the study without concentrating on building a connection between this group's ecological importance and its phylogeny. In specific, the abstract and introduction as it is written appear as a series of random facts put together. There must be some level of connectivity between your sentences, building a storyline that leads your readership to grasp the importance of your study. This is missing and would be essential as your findings are important. In the discussion, some paragraphs are placed without explaining how they relate to this study or its findings. Please, relay these to a broad readership. Particularly, please pay attention to the following:

1.      Sentences in each paragraph must follow the same storyline. A connectivity is necessary.

2.      Using past tense or present tense and not both.

3.      A precise choice of wording that corresponds to your findings.

4.      New concepts, metrics, and/or analyses cannot be introduced in the result or discussion sections. They must first be introduced in the methodology with proper citation.

5.      A proper citation must be provided if the comparison is made with other studies or insect taxa. Otherwise, the readership doesn’t know what is compared to what.

6.      The species' scientific name must be written in italics, full, with the author and date. After that, they can be abbreviated. Since this is a systematics paper, it is very important. Please go through the script and correct this.

Please see my further comments in the manuscript and respond and correct accordingly.

Author Response

Dear reviewer 1,

On behalf of my co-authors, we thank you very much for giving us an opportunity to revise our manuscript. We have tried our best to revise our manuscript according to the comments. The main corrections in the paper and the responds to your comments are as flowing.

Comments 1: The authors diligently have worked on writing this article, which has yielded results that are not easily obtained, and I commend them. The use of mitogenome for constructing the phylogenetic of Chironomidae species is a new approach, and I am eager to see this study in publication as the information it generates is essential for future studies. However, I would like to raise a point of consideration that may question their entire outcome. The results obtained in this study could potentially be influenced by the outgroup choice, which is from the same generic complex. While I understand that the number of species with full mitogenome sequence is limited, there are still numerous other genera (see Zhang et al. 2023, for instance) that could have been chosen as an outgroup. The authors do not convincingly present an argument for their choice of outgroup. The choice of the outgroup is based on a more distantly related group of organisms, usually in other subfamilies, tribes, or species groups. Please either provide a convincing and comprehensive reason for your choice of the outgroup or redo the phylogenetic analysis with a different outgroup(s). Have the authors done other analyses with different outgroups? If so, please provide those. If choosing other outgroups (e.g., Polypedilum yongsanensis or Endochironomus albipennis or both) over the present outgroups do not change the relationship, then your outgroup choice is still valid.

Response 1:We appreciate very much for your positive and constructive comments and suggestions on our manuscript. We agree with this comment. Our article primarily focuses on reporting the mitochondrial genome data of Harnischia cpx and discussing the relationships between three genera that exhibit close morphological similarities and cluster together in Saether's (1977) classification, rather than exploring the internal relationships within the Harnischia complex. Consequently, we selected two other closely related genera as outgroups.

Comments 2: In terms of writing, there seems to be a level of haste in getting to the phylogenetic part of the study without concentrating on building a connection between this group's ecological importance and its phylogeny. In specific, the abstract and introduction as it is written appear as a series of random facts put together. There must be some level of connectivity between your sentences, building a storyline that leads your readership to grasp the importance of your study. This is missing and would be essential as your findings are important. In the discussion, some paragraphs are placed without explaining how they relate to this study or its findings.

Response 2:Thank you for your patient advice. We have reorganized the language and rewritten the text, emphasizing the ecological significance and systematic relationships. All modifications, including those in the abstract, introduction, and discussion sections, have been highlighted in red within the article.

Comments 3: Sentences in each paragraph must follow the same storyline. A connectivity is necessary.

Response 3:Thank you for your suggestions. We have made the necessary revisions and highlighted them in red.

Comments 4: Using past tense or present tense and not both.

Response 4: Thank you for your valuable advice. We have standardized the tense throughout the document to the past tense.

Comments 5: A precise choice of wording that corresponds to your findings.

Response 5:Thank you for your suggestions. We have made the necessary revisions and highlighted them in red.

Comments 6: New concepts, metrics, and/or analyses cannot be introduced in the result or discussion sections. They must first be introduced in the methodology with proper citation.

Response 6: We have supplemented the Methods section with additional relevant information and included pertinent citations.

Comments 7: A proper citation must be provided if the comparison is made with other studies or insect taxa. Otherwise, the readership doesn’t know what is compared to what.

Response 7:We have incorporated pertinent citations accordingly.

Comments 8: The species' scientific name must be written in italics, full, with the author and date. After that, they can be abbreviated. Since this is a systematics paper, it is very important. Please go through the script and correct this.

Response 8:We have revised and completed the scientific names in the main text to ensure their accuracy and completeness.

We appreciate for your warm work earnestly and hope that the correction will meet with approval.

Once again, thank you very much for your comments and suggestions.

Wenbin Liu

Reviewer 2 Report

Comments and Suggestions for Authors

Dear editor and authors,

The paper has a overall high quality and brings interesting molecular analyses. I recommend the publication but after considering my suggestions on the outgroup selection (see file attached) among others.

Congratulation for your research!

With kindest regards

Author Response

Dear reviewer 2,

On behalf of my co-authors, we thank you very much for giving us an opportunity to revise our manuscript. We have tried our best to revise our manuscript according to the comments. The main corrections in the paper and the responds to your comments are as flowing.

Comments 1: The paper has a overall high quality and brings interesting molecular analyses. I recommend the publication but after considering my suggestions on the outgroup selection (see file attached) among others.

Response 1:We appreciate very much for your positive and constructive comments and suggestions on our manuscript. We agree with this comment. Our article primarily focuses on reporting the mitochondrial genome data of Harnischia cpx and discussing the relationships between three genera that exhibit close morphological similarities and cluster together in Saether's (1977) classification, rather than exploring the internal relationships within the Harnischia complex. Consequently, we selected two other closely related genera as outgroups.

We appreciate for your warm work earnestly and hope that the correction will meet with approval.

Once again, thank you very much for your comments and suggestions.

Wenbin Liu

Round 2

Reviewer 2 Report

Comments and Suggestions for Authors

Dear authors and editor,

All suggestions regarding the problems on the methods and conclusions of phylogenetic analysis were considered and overall seems fine although no outgroup selection is mentioned. Even though I am considering to accept to publish, I strongly suggest the editor to look for another reviewer with great expertise in phylogenetic analysis.

With kindest regards